# The intellectual disability gene Kirrel3 regulates target-specific mossy fiber synapse development in the hippocampus

**E Anne Martin[1†], Shruti Muralidhar[1†], Zhirong Wang[1], Diégo Cordero Cervantes[1], Raunak Basu[1], Matthew R Taylor[1], Jennifer Hunter[1], Tyler Cutforth[2], Scott A Wilke[3], Anirvan Ghosh[4], Megan E Williams[1*]**

[1]Department of Neurobiology and Anatomy, University of Utah School of Medicine, Salt Lake City, United States; [2]Department of Neurology, Columbia University, New York City, United States; [3]Neurobiology Section, Division of Biological Sciences, University of California, San Diego, San Diego, United States; [4]Neuroscience Discovery, Roche Innovation Center Basel, F. Hoffmann-La Roche, Basel, Switzerland

**Abstract** Synaptic target specificity, whereby neurons make distinct types of synapses with different target cells, is critical for brain function, yet the mechanisms driving it are poorly understood. In this study, we demonstrate Kirrel3 regulates target-specific synapse formation at hippocampal mossy fiber (MF) synapses, which connect dentate granule (DG) neurons to both CA3 and GABAergic neurons. Here, we show Kirrel3 is required for formation of MF filopodia; the structures that give rise to DG-GABA synapses and that regulate feed-forward inhibition of CA3 neurons. Consequently, loss of Kirrel3 robustly increases CA3 neuron activity in developing mice. Alterations in the Kirrel3 gene are repeatedly associated with intellectual disabilities, but the role of Kirrel3 at synapses remained largely unknown. Our findings demonstrate that subtle synaptic changes during development impact circuit function and provide the first insight toward understanding the cellular basis of Kirrel3-dependent neurodevelopmental disorders.

**\*For correspondence:** megan. williams@neuro.utah.edu

[†]These authors contributed equally to this work

**Competing interests:** The authors declare that no competing interests exist.

## Introduction

Executing cognitive tasks requires coordination among neural circuits. Therefore, neurons usually send and receive neural information with different synaptic partners. One way neurons differentially regulate activity among partners is by forming different types of synapses with each partner (*Williams et al., 2010*; *Emes and Grant, 2012*). This kind of synaptic target specificity is exquisitely exemplified by hippocampal mossy fiber (MF) synapses. MF synapses connect glutamatergic dentate granule (DG) neurons to glutamatergic CA3 neurons and GABAergic interneurons (GABA neurons). The main DG-CA3 synapse consists of a giant presynaptic bouton apposed to a multi-headed CA3 spine called a thorny excrescence (TE). In addition, filopodia project from the main bouton and synapse with nearby GABA neurons (*Frotscher, 1989*; *Acsády et al., 1998*). Filopodial MF synapses mediate feed-forward inhibition of CA3 neurons and are essential for hippocampal function during learning and memory tasks (*Torborg et al., 2010*; *Ruediger et al., 2011*). Although main bouton and filopodial MF synapses are physically linked, they have different molecular and functional properties (*Toth et al., 2000*; *McBain, 2008*). This suggests DG neurons utilize specific cues to construct different types of synapses with CA3 and GABA neurons, but the identity of the target-specific cues is unknown.

**eLife digest** Nerve cells in the brain connect to each other via junctions called synapses to form vast networks that process information. Much like streets can be joined with stop signs, traffic lights, or exit ramps depending on the flow of traffic, different types of synapses control the flow of information along nerves in distinct ways.

In a region of the brain called the hippocampus, nerve cells called DG neurons are connected to other neurons by two different types of synapses. One type of synapse allows the DG neurons to activate CA3 neurons, while the second type allows the DG neurons to activate GABAergic neurons. These same GABAergic neurons can then inhibit the activity of the CA3 neurons. Therefore, through these two different types of synapses, DG neurons can both increase and decrease the activity of the CA3 neurons. This delicate balance of activity across the two types of DG synapses is very important for the hippocampus to work properly, which is critical for our ability to learn and remember.

Mutations in the gene that encodes a protein called Kirrel3 are associated with autism, Jacobsen's syndrome, and other disorders that affect intellectual ability in humans. Kirrel3 is similar to a protein found in roundworms that regulates the formation of synapses, but it is not known if it plays the same role in humans and other mammals. Now, Martin, Muralidhar et al. studied the role of Kirrel3 in mice.

The experiments show that Kirrel3 is produced in both the DG neurons and the GABAergic neurons, but not the CA3 neurons. Young mutant mice that lacked Kirrel3 made fewer synapse-forming structures between DG neurons and GABAergic neurons than normal mice, but the synapses that connect DG neurons to CA3 neurons formed normally. This disrupted the balance of activity across the two types of DG synapses and the CA3 neurons in the mutant mice were over-active.

Together, Martin, Muralidhar et al.'s findings show that altering the levels of Kirrel3 can alter the balance of synapses in the hippocampus. This may explain how even very small changes in synapse formation during brain development can have a big impact on nerve cell activity. The next challenge is to understand exactly how Kirrel3 helps build synapses, which may lead to the development of new drugs that help to rebalance brain activity in people that lack Kirrel3.

Kirrel1, 2, and 3 (also known as Neph1, 3, and 2, respectively) are transmembrane immunoglobulin superfamily members (*Figure 1D*). In mice, Kirrels have been studied during formation of the kidney slit diaphragm, an adhesive cell junction for filtering blood (*Donoviel et al., 2001*; *George and Holzman, 2012*) and in axon targeting of the olfactory and vomeronasal systems (*Serizawa et al., 2006*; *Prince et al., 2013*). In humans, copy number variations and exonic point mutations in the Kirrel3 gene are associated with intellectual disability, autism, and Jacobsen's syndrome, a rare developmental disorder that often includes intellectual disabilities (*Bhalla et al., 2008*; *Guerin et al., 2012*; *Michaelson et al., 2012*; *Neale et al., 2012*). The Kirrel ortholog SYG-1 regulates synapse formation and axon branching in *Caenorhabditis elegans* (*Shen and Bargmann, 2003*; *Chia et al., 2014*), but the role of Kirrel3 in mammalian synapse development is unknown. Here, we demonstrate Kirrel3 is a target-specific cue at MF synapses. Kirrel3 specifically regulates development of DG-GABA MF filopodia, which are necessary to constrain excitatory drive to CA3 neurons after DG stimulation.

## Results

### Kirrel3 is a homophilic, synaptic adhesion molecule

Given the association between Kirrel3 mutations and intellectual disabilities, we investigated the role of Kirrel3 in hippocampal circuits, which are critical for learning and memory, and may be impaired in patients with intellectual disabilities. Kirrel3 protein is enriched in synaptosomes prepared from hippocampal lysates with greatest enrichment at postnatal day (P) 21 (*Figure 1A*). Next, we obtained Kirrel3 knockout mice, which were recently described (*Prince et al., 2013*) and prepared

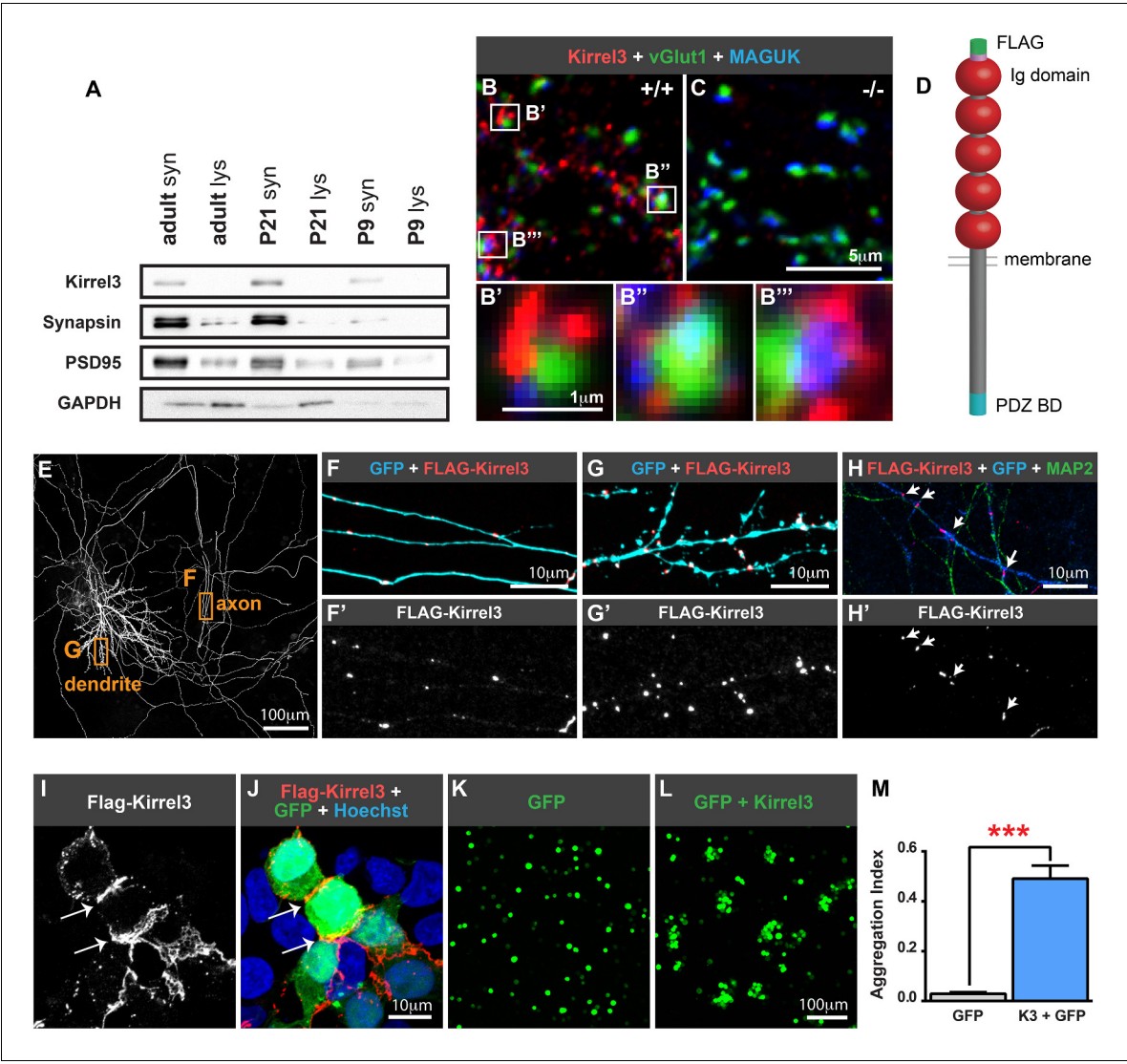

**Figure 1.** Kirrel3 is a synaptic molecule that mediates homophilic, trans-cellular adhesion. (A) Synaptosomes from mouse hippocampi from P9, P21, and adult (P55) were immunoblotted for indicated proteins. lys; lysate. syn; synaptosome. 2 μg protein per lane. (B, C) 14 days in vitro (DIV) cultured hippocampal neurons immunostained with antibodies against Kirrel3 (red), vGlut1 (green), and MAGUK proteins (blue). Boxed regions in B are magnified below in B', B'', and B'''. Neurons from Kirrel3 knockout mice have no Kirrel3 signal (C). (D) Diagram of Kirrel3 protein and location of inserted FLAG tag. Ig; immunoglobulin. (E–H) Cultured hippocampal neurons were co-transfected with FLAG-Kirrel3 and GFP and immunostained for indicated proteins. Anti-FLAG antibodies were added prior to fixation to label only surface Kirrel3. Note surface Kirrel3 is seen as puncta on axons and dendrites after synapse formation in 14DIV neurons (E–G) and prior to synapse formation in 4DIV neurons (H). H shows that surface Kirrel3 also clusters at axon–dendrite crossings. Boxed regions in E are shown magnified in F and G. FLAG signal alone is shown in lower panels F', G', and H'. (I, J) Kirrel3 clusters at cell junctions. 293HEK cells were co-transfected with GFP and FLAG-Kirrel3, immunostained for GFP and FLAG, and nuclei labeled with Hoechst. (K–M) CHO cells transfected with either GFP control or GFP and Kirrel3 were tested for adhesion. Only cells expressing Kirrel3 formed aggregates. Aggregation index was calculated by dividing the total GFP fluorescence in cell aggregates by the total GFP fluorescence in the well. Mean ± SEM are shown, n = 3, *** indicates p = 0.001 by two-tailed t-test.

The following figure supplements are available for figure 1:

**Figure supplement 1.** Kirrel1 and 2 are not expressed in the hippocampus.

**Figure supplement 2.** Kirrel3 undergoes homophilic binding in cells.

hippocampal neuron cultures from newborn wild-type and knockout mice. In hippocampal neurons cultured for 14 days in vitro (DIV), Kirrel3 localizes to puncta adjacent to the pre- and post-synaptic markers vGlut1 and MAGUK in wild-type but not knockout neurons (*Figure 1B,C*). This suggests that, like cadherins, Kirrel3 localizes to perisynaptic adhesion zones rather than the synaptic cleft. To determine if Kirrel3 is axonal, dendritic, or both, we analyzed the distribution of surface-expressed FLAG-Kirrel3 (*Figure 1D*) in sparsely transfected neurons using live labeling. Surface FLAG-Kirrel3 is seen as puncta on axons and dendrites of 14DIV neurons (*Figure 1E–G*). Moreover, even prior to synaptogenesis in 4DIV neurons, FLAG-Kirrel3 already has a punctate distribution in axons and dendrites and clusters at axon–dendrite contact points (*Figure 1H*).

Kirrels can function via homophilic binding or heterophilic binding to nephrin, another Ig superfamily member (*Gerke et al., 2003*; *Serizawa et al., 2006*). However, neither nephrin nor the other Kirrel family members, Kirrel1 and Kirrel2, have appreciable expression in the hippocampus (*Putaala et al., 2001*) (*Figure 1—figure supplement 1*). We noticed Kirrel3 clusters at cell junctions (*Figure 1I,J*) and therefore we directly tested the adhesive ability of Kirrel3 homophilic interactions using a cell aggregation assay. We demonstrate Kirrel3 mediates trans-cellular homophilic binding (*Figure 1K–M* and *Figure 1—figure supplement 2*). Taken together, our data indicate Kirrel3 is present at early axon–dendrite contacts, localizes at or near synapses, and is a bona fide homophilic adhesion molecule, all of which implicate Kirrel3 in synapse development.

## Hippocampal DG and GABA neurons express Kirrel3

Next, we determined which hippocampal neurons express Kirrel3. In developing P14 and adult hippocampi, Kirrel3 mRNA is present in two cell types: (1) DG neurons and (2) scattered cells of the hilus and area CA3 (*Figure 2A–D* and *Figure 2—figure supplement 1A*). Correspondingly, Kirrel3 protein is present in the molecular and stratum lucidum layers of the hippocampus, containing DG dendrites and axons, respectively (*Figure 2E*). It is also present in scattered cells of the hilus and area CA3 (*Figure 2F*) and faintly in the stratum lacunosum-moleculare, which contains axons from entorhinal cortex. No Kirrel3 signal was detected in knockout mice (*Figure 2G,H* and *Figure 2—figure supplement 1B*). Instead, Kirrel3 knockout mice have farnesylated GFP in frame after exon 1 so they express membrane-associated GFP instead of Kirrel3 (*Prince et al., 2013*). Examination of GFP expression in knockout mice indicates that again, Kirrel3 is selectively expressed by DG neurons and scattered cells of area CA3 (*Figure 2I–O*). Notably, we never observe GFP expression in CA3 neurons (*Figure 2—figure supplement 1C–E*).

The scattered Kirrel3-positive cells reside mainly outside the pyramidal layer and have GFP-labeled arbors. This suggests they may be GABAergic interneurons. To test this, we co-stained P14 Kirrel3 heterozygous and knockout mice with antibodies against GFP to identify Kirrel3-expressing cells and GABA to identify GABAergic interneurons (*Figure 2I–S* and *Figure 2—figure supplement 2*). We find that nearly all Kirrel3-expressing cells express GABA (*Figure 2S*). Conversely, about 20% of all GABA neurons in area CA3 express Kirrel3 (*Figure 2S*). We also co-stained with several common GABA neuron markers and find that two thirds (about 67%) of Kirrel3/GABA neurons express calbindin (*Figure 2P–S*) and these Kirrel3-positive neurons make up about half of all calbindin-positive interneurons. Notably, glutamatergic DG neurons also highly express Kirrel3 and calbindin, providing a shared molecular profile between these neuronal populations. Kirrel3 cell populations are similar in P14 Kirrel3 heterozygous and knockout mice (*Figure 2—figure supplement 2*), suggesting that complete loss of Kirrel3 does not change cell fate or induce cell death of the neurons examined. Taken together, we find that, in the hippocampus, DG neurons and a subset of calbindin-positive GABA neurons selectively express Kirrel3.

## Kirrel3 regulates MF filopodia development

Because Kirrel3 is a homophilic molecule expressed by DG and GABA neurons, we hypothesized Kirrel3 homophilic interactions may specifically regulate formation of MF filopodia connecting DG axons to GABA dendrites during development (*Figure 3A*). To test this, we analyzed MF presynaptic morphology in Kirrel3 wild-type and knockout mice at P14, the peak of MF synaptogenesis. DiI crystals were placed in the DG cell body layer of fixed brains. After 1 week, dye diffuses along DG axons and labels MF presynaptic terminals (*Figure 3B–G*). We discovered Kirrel3 knockout mice have significantly fewer and shorter filopodia than wild-type (*Figure 3B–M* and *Figure 3—figure*

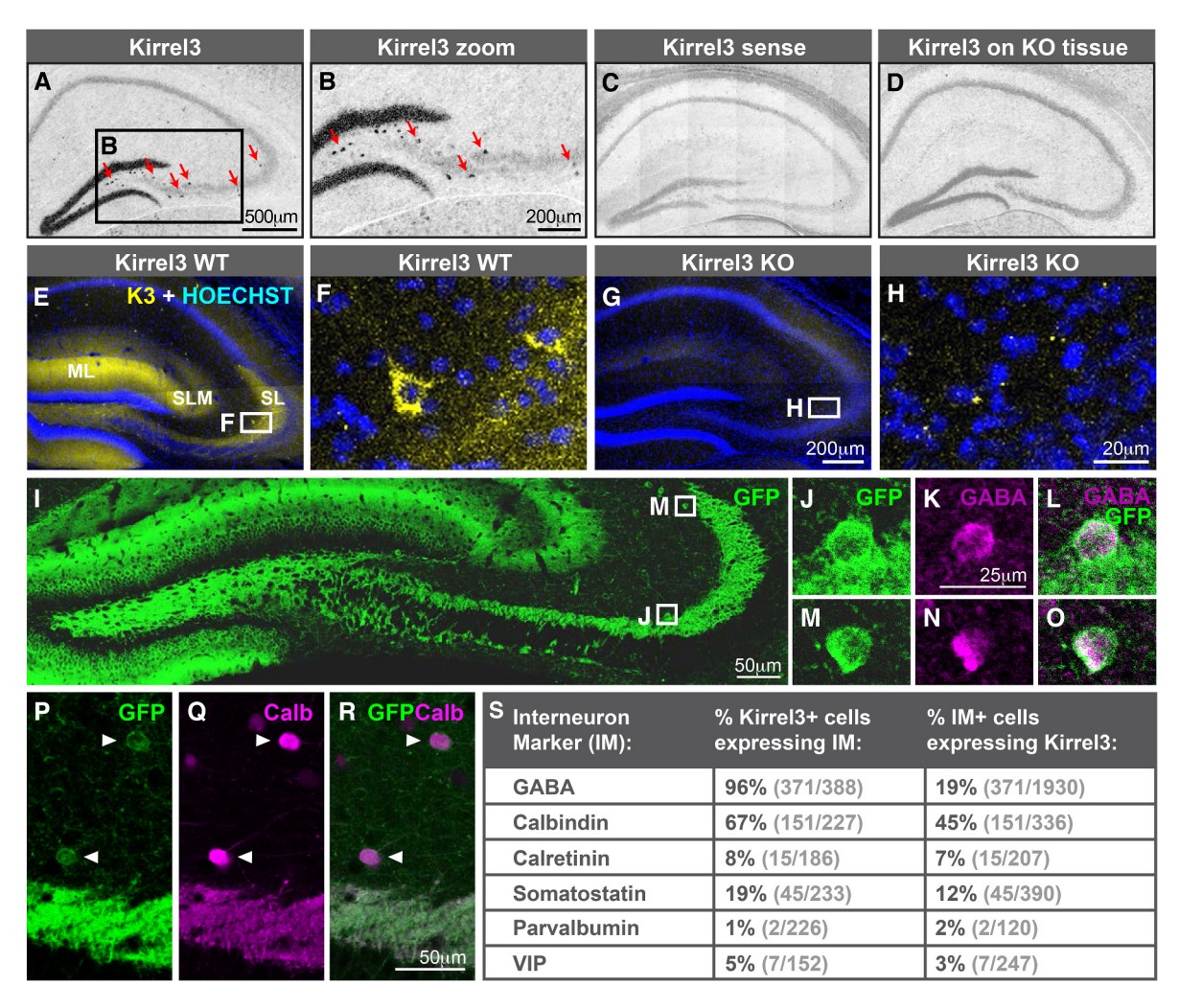

**Figure 2.** Hippocampal DG and GABA neurons express Kirrel3. (**A–D**) In situ hybridizations for Kirrel3 mRNA on adult P60–P70 hippocampal sections from WT (**A–C**) and KO (**D**) mice. A negative control sense probe on WT tissue is shown in **C**. Red arrows in boxed region of **A** point to scattered Kirrel3-expressing cells shown magnified in (**B**). (**E–H**) Hippocampal sections from Kirrel3 WT (**E, F**) and KO (**G, H**) mice were immunostained with anti-Kirrel3 antibodies (yellow) and Hoechst (blue). **F** and **H** are magnified images of boxed regions in **E** and **G**. (**I–O**) P14 Kirrel3 KO mice with farnesylated GFP inserted in the Kirrel3 locus were immunostained with anti-GFP antibodies to identify Kirrel3-expressing cells (green). Dentate granule (DG) dendrites and their mossy fiber (MF) axons are brightly labeled (**I**) as well as GABA-expressing cells (magenta) in area CA3. (**P–R**) P14 Kirrel3 KO mice were immunostained for GFP (green) and calbindin (Calb, magenta). (**S**) Analysis of Kirrel3-positive cells in P14 Kirrel3 KO mice co-expressing interneuron markers. Abbreviations: wild-type, WT; knockout, KO; molecular layer, ML; stratum lucidum, SL. Stratum lacunosum-moleculare, SLM. Images in **A–D**, **E**, **G**, and **I** are tiled.

The following figure supplements are available for figure 2:

**Figure supplement 1.** Kirrel3 is not expressed by CA3 neurons.

**Figure supplement 2.** Kirrel3 is expressed by mainly calbindin-positive GABA neurons.

supplement 1). In contrast, the main bouton area and perimeter are not affected by loss of Kirrel3 (*Figure 3J,K*). To examine the main synapse in more detail, we analyzed postsynaptic CA3 TE spines by filling CA3 neurons with Alexa568 dye (*Figure 3N,O*). TE spine head density and length are similar between P21 Kirrel3 wild-type and knockout mice (*Figure 3P,Q*). Thus, Kirrel3 knockout mice have significant morphological defects in MF filopodia but not in pre- or post-synaptic structures of

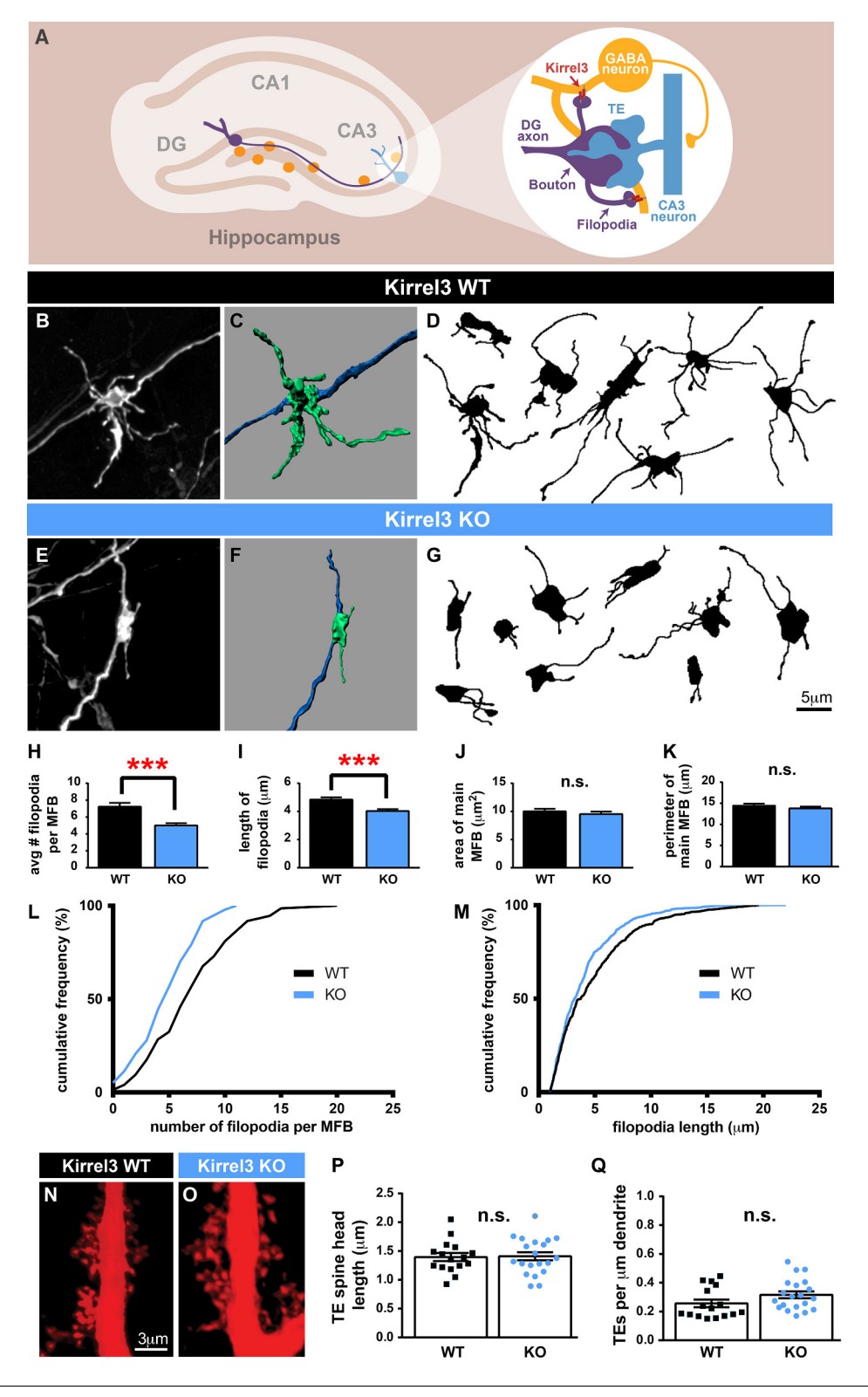

**Figure 3.** Kirrel3 regulates MF synapse form and function during development. (**A**) MF synapse diagram. TE; thorny excrescence. (**B, E**) DiI-labeled MF synapses from P14 Kirrel3 WT and KO mice. (**C, F**) 3D renderings of synapses in (**B, E**). (**D, G**) Tracings of representative DiI-labeled MF synapses. (**H–M**) MF synapse morphology quantification. The number of filopodia per MF bouton (**H, L**) and filopodia length (**I, M**) are reduced in Kirrel3 KO

*Figure 3 continued on next page*

*Figure 3 continued*

mice. L and M are cumulative histograms of data shown in H and I, respectively. Area (J) and perimeter (K) of the main MF bouton are unaffected by genotype. n = 74 WT and 97 KO MF synapses from four mice of each genotype. Two-tailed t-tests: in H, *** = $p < 0.001$ and in I, *** = $p = 0.0001$. (N, O) Examples of P21 CA3 TE spines labeled by iontophoresis. (P, Q) Quantification of P21 spine morphology. No significant differences as determined by two-tailed t-tests. n = 16 WT neurons from four animals and 20 KO neurons from three animals. All bar graphs show mean ± SEM.

The following figure supplement is available for figure 3:

**Figure supplement 1.** Kirrel3 is required for normal development of MF filopodia.

main MF synapses, suggesting that the number of DG synapses onto GABA neurons is reduced in Kirrel3 knockout mice.

## Kirrel3 regulates CA3 neuron excitability driven by MF input

If Kirrel3 knockout mice have fewer DG-GABA synapses, there should be reduced excitation to GABA neurons, particularly to Kirrel3-positive GABA neurons. Unfortunately, Kirrel3-positive GABA neurons cannot be identified in wild-type mice making it impossible to target them for direct electrophysiological studies at this time. However, reducing DG-GABA synapses in area CA3 is expected to decrease feed-forward inhibition and thereby increase excitation of CA3 neurons after DG stimulation. To test this, we recorded excitatory and inhibitory currents evoked in CA3 neurons after DG stimulation in acute slices (*Figure 4A*). As predicted by our model, P14–P16 Kirrel3 knock-out mice have a significantly increased excitatory/inhibitory (E/I) ratio compared to wild-type mice (*Figure 4B*). We confirmed evoked responses were due to MF stimulation by applying DCG-IV, a metabotropic glutamate receptor agonist that selectively inhibits MF release (*Figure 4A*) (*Kamiya et al., 1996*; *Yoshino et al., 1996*; *Torborg et al., 2010*).

To test if Kirrel3 knockout mice have increased CA3 activity in an awake and behaving animal, we analyzed cFos, an immediate early gene marking recently activated neurons (*Kawashima et al., 2014*), in P14 Kirrel3 wild-type and knockout mice. Mice were either removed from the home cage for immediate fixation (no stim) or allowed to explore an enriched environment for 25 min prior to fixation (stim). At this age, basal, unstimulated cFos expression is low regardless of genotype, but with stimulation, P14 knockout mice have a twofold increase in cFos-positive CA3 neurons compared to wild-type (*Figure 4C,D*). As a control, mice were recorded and exploration was comparable between genotypes (*Figure 4—figure supplement 1*). Interestingly, the number of cFos-positive neurons did not significantly increase in wild-type mice after stimulation but this is consistent with previous reports (*Waters et al., 1997*) and likely reflects that at P14 the CA3 is dominated by inhibition (*Figure 4B*, note wild-type E/I ratio is <1) (*Torborg et al., 2010*). Because Kirrel3 is expressed by entorhinal cortex axons that innervate the hippocampus, we also generally examined inputs to CA3 and DG neurons by recording miniature excitatory postsynaptic currents (mEPSCs). No significant differences between wild-type and Kirrel3 knockout neurons in mEPSC amplitude or frequency were found for either cell type (*Figure 4—figure supplement 1*). Together, our functional data support the hypothesis that Kirrel3 selectively regulates formation of filopodial DG-GABA MF synapses.

## Morphological defects in MF synapses persist in Kirrel3 knockout mice

MF synapse complexity peaks at P14 and is refined such that adult synapses have fewer filopodia and a larger main bouton (*Wilke et al., 2013*). To investigate MF synapse maturation under sustained loss of Kirrel3, we analyzed MF morphology in adult (P60–P75) wild-type and knockout mice. Both genotypes have fewer filopodia as adults than at P14 (compare *Figure 4E* and *Figure 3H*), suggesting age-dependent filopodia refinement is Kirrel3-independent. However, while P14 knockout mice have fewer filopodia with a normal main bouton, adult knockout mice have fewer filopodia and a smaller main bouton (*Figure 4E–J*). Thus, MF synapse defects worsen with age in Kirrel3 knockout mice. It is possible Kirrel3 is directly required for main bouton maturation or maintenance, but Kirrel3 is not expressed by CA3 neurons. Another possibility is that defects in the main bouton in adult knockout mice result from compensatory mechanisms enacted to dampen the dramatic increase in

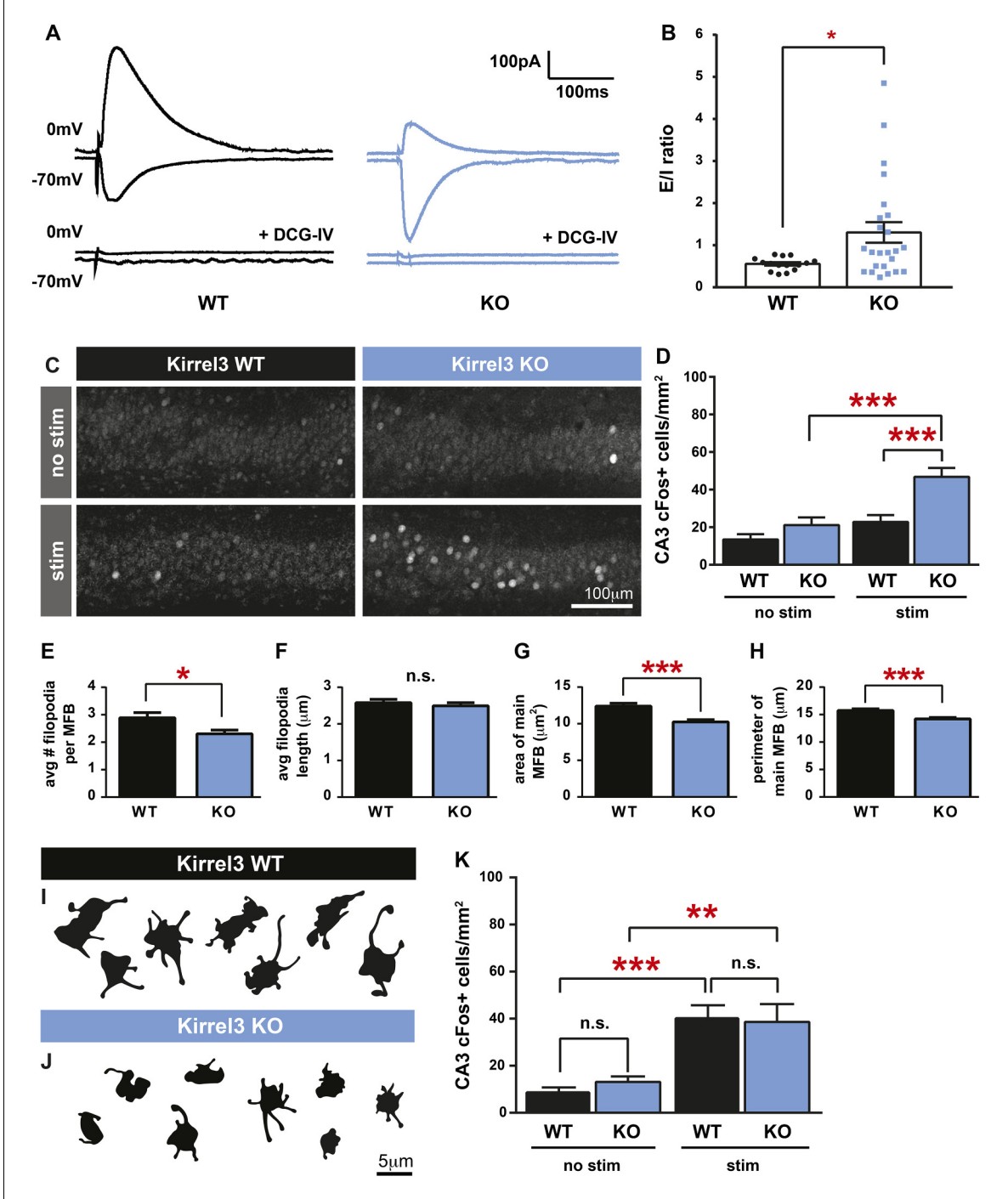

**Figure 4.** Kirrel3 regulates the activity of CA3 neurons during development. (A) Evoked responses at −70 and 0 mV of a CA3 neuron after stimulation of the MF pathway in Kirrel3 WT and KO mice. Lower traces show responses of the same cells after perfusion of 0.5 μm DCG-IV. (B) Average excitatory/inhibitory (E/I) ratio for CA3 neurons recorded from P14–P16 WT and KO mice. n = 15 cells from five WT animals and 24 cells from five KO animals. p = 0.02 with unpaired t-test. (C) Examples of anti-cFos staining in CA3 neurons from P14 mice. Note increased cell staining in Kirrel3 KO mice after 25 min stimulation (stim) in a novel, enriched environment. (D) Quantification of cFos-positive CA3 neurons at P14. n = 14 (WT no stim), 15 (WT stim), 15 (KO no stim), and 15 (KO stim) sections from three mice per condition. Two-way ANOVA indicates there is a significant difference among condition (no stim vs stim) and genotype. p values from post-tests are 0.0001 (KO no stim vs KO stim) and 0.0004 (WT stim vs KO stim). (E–H) Quantification of MF synapse structure in adult mice. n = 115 WT and 131 KO synapses from three mice per genotype. Two-tailed t-tests indicate p = 0.01 (E), p < 0.0001 (G), and p = 0.002 (H). (I, J) Tracings of representative DiI-labeled MF synapses from adult (P60–P75) Kirrel3 WT and KO mice. (K) Quantification of cFos-positive cells in area CA3 of adult mice. n = 14 (WT no stim), 14 (WT stim), 16 (KO no stim), and 16 (KO stim) sections from three different mice per condition.
*Figure 4 continued on next page*

*Figure 4 continued*

Two-way ANOVA indicates there is a significant difference among condition (no stim vs stim) but not genotype. p values from post-tests are 0.0005 (WT no stim vs WT stim) and 0.003 (KO no stim vs KO stim). All graphs show mean ± SEM.
The following figure supplement is available for figure 4:

**Figure supplement 1.** Spontaneous mEPSC activity of CA3 and DG neurons is normal in the absence of Kirrel3.

CA3 neuron excitability during development. To test this, we analyzed cFos expression in adult mice to determine if knockout CA3 neuron activity returns to normal. In support, unstimulated and stimulated cFos expression in CA3 neurons is similar between P60 adult wild-type and knockout mice (*Figure 4K* and *Figure 4—figure supplement 1*). Thus, our data suggest a model in which a reduction in MF filopodia during development leads to hyper-activation of CA3 neurons that, with time, causes homeostatic mechanisms to decrease the size of the main DG-CA3 MF synapse and return overall CA3 activity levels to a set point. This model remains to be tested in more detail in future studies. Interestingly, a battery of behavioral tests was recently performed on adult Kirrel3 knockout mice (*Choi et al., 2015*). Kirrel3 knockout mice have only mild hyperactivity and a moderate impairment in novel object recognition compared to wild-type (*Choi et al., 2015*). Our results indicate CA3 neuron activity is significantly impaired in 14-day-old animals but returns to normal by 2 months of age. This may explain why adult Kirrel3 knockout mice have only moderate behavioral impairments and suggests that younger animals may have more severe behavioral problems.

## Discussion

In summary, we demonstrate Kirrel3 is required for normal development of MF filopodia, the synaptic structures connecting DG and GABA neurons. Because Kirrel3 is a homophilic adhesion molecule expressed by DG neurons and calbindin-positive GABA neurons in hippocampal area CA3, our work suggests trans-cellular Kirrel3 interactions may stabilize MF filopodia contact and subsequent synapse formation between Kirrel3-expressing cells. Functionally, we demonstrate Kirrel3 is required to maintain feed-forward inhibition and constrain CA3 neuron activity in young animals. Although a few molecules have been shown to generally affect MF presynapse formation (*Danzer et al., 2008*; *Williams et al., 2011*; *Lanore et al., 2012*; *Wilke et al., 2012*), Kirrel3 is among the first shown to selectively act on MF filopodia and our results provide evidence that main and filopodial MF synapse formation can be uncoupled. The role of calbindin-positive GABA neurons in the hippocampus is little studied, but our findings suggest they may receive a substantial fraction of DG MF input and be a critical component of feed-forward inhibitory circuits regulating CA3 activity.

Given that alterations in the Kirrel3 gene are associated with autism and intellectual disabilities, this work provides the first insight into cellular mechanisms that may underlie Kirrel3-dependent neurological disorders. Our work suggests Kirrel3 loss selectively reduces excitatory synapses made onto inhibitory neurons in hippocampal area CA3. Consequently, CA3 neurons are over-active in young Kirrel3 knockout animals. Altered E/I ratios are hypothesized to underlie neurodevelopmental disorders (*Rubenstein and Merzenich, 2003*; *Baroncelli et al., 2011*; *Fakhoury, 2015*), and our work suggests this may be an important circuit defect of Kirrel3-associated diseases. Interestingly, such a strong imbalance probably would not occur if all synapses were equally impaired. In this way, loss of synaptic specificity molecules can cause widespread circuit dysfunction. There is much evidence that hippocampal circuits are altered in autism and intellectual disabilities, but it is likely not the only brain region affected in these complex neurological disorders (*Philip et al., 2012*; *Zoghbi and Bear, 2012*). Kirrel3 is also expressed by specific populations of neurons outside the hippocampus (*Lein et al., 2007*; *Choi et al., 2015*). Thus, Kirrel3 may also regulate formation of other specific types of synapses throughout the brain, which may further contribute to its association with neurodevelopmental disorders.

## Materials and methods

### Mouse lines

Generation of the Kirrel3 knockout mouse line was recently reported (*Prince et al., 2013*). All animals and experiments were maintained and conducted in accordance with the NIH guidelines on the care and use of animals and approved by the University of Utah IACUC committee.

### Kirrel3 cloning and RNA in situ hybridization

*Kirrel3* cDNA was kindly provided by Dr Hitoshi Sakano and *Kirrel1* and *Kirrel2* cDNAs were obtained from OpenBiosystems. Standard PCR cloning was used to move cDNAs to the pBos vector and add an extracellular FLAG tag after the signal sequence. Full-length *Kirrel1*, *2*, and *3* cDNAs were used to generate sense and anti-sense probes. Standard DIG-labeled, non-radioactive in situ hybridization protocol was carried out using the Roche DIG-labeling kit on coronal cryosections of brain tissue.

### Cell culture

Neurons: P0 rat cortical glia were cultured on PDL/collagen-coated coverslips to form a monolayer. 1 week later, P0 mouse or rat hippocampi were dissected in cold 4-(2-hydroxyethyl)-1-piperazinee-thanesulfonic acid (HEPES)-buffered saline solution, incubated in papain for 30 min, dissociated, and plated to glial monolayers at $4$–$5 \times 10^4$ cells/ml. All media was from Life Technologies (Carlsbad, CA, United States). Glia media: DMEM, 10% Fetal Bovine Serum (FBS), 75 mM glucose, and penicillin/streptomycin. Neuron-plating media: MEM, 10% horse serum, 50 mM glucose, 0.250 mM pyruvic acid, 2 mM Glutamax, 100 U/ml penicillin, 100 µg/ml streptomycin. Neuron-feeding media: Neurobasal A, B27, 30 mM glucose, 0.5 mM Glutamax, 20 U/ml penicillin, 20 µg/ml streptomycin. Neuron transfections were done using the calcium-phosphate method (*Dudek et al., 2001*). Cell lines: 293HEK media: DMEM, 10% FBS, and penicillin/streptomycin. CHO media: F12K media, 10% FBS, and penicillin/streptomycin. Cell line transfections were done using polyethylenimine (PEI, Polysciences, Warrington, PA, United States) at a ratio of 5 µg PEI/1 µg DNA.

### CHO cell aggregation assay

CHO cells were co-transfected with FLAG-Kirrel3 pBOS (4 µg) and GFP pBOS (2 µg) using PEI. 48 hr later, cells were washed with HEPES $Mg^{2+}$ free (HMF) buffer (137 mM NaCl, 5.4 mM KCl, 1 mM $CaCl_2$, 0.34 mM $Na_2HPO_4$, 5.5 mM glucose, 10 mM HEPES, pH 7.4 adjusted with NaOH) and detached from the dishes using 0.01% trypsin in HMF. Detached cells were spun down, resuspended in HMF, counted, and 100,000 cells were pipetted into single wells of 24-well plates precoated with 1% BSA in HMF. Subsequently, the plates were placed on a nutator for 90 min at 37°C. The cells were then fixed with paraformaldehyde (PFA) (4% final concentration), transferred to a 96-well glass bottom plate, and imaged in a Zeiss LSM 710 confocal microscope. The aggregation index was calculated by dividing the total GFP fluorescence in cell aggregates by the total GFP fluorescence in the well. Analysis was done using ImageJ.

### Immunostaining

Cultured cells were fixed in 4% PFA for 10 min, washed with phosphate-buffered saline (PBS), and incubated in blocking solution (PBS with 3% bovine albumin and 0.1% Triton-X100) for 30 min. Primary antibody was diluted in blocking solution and incubated on cells for 1–2 hr. After three washes, secondary antibody was incubated for 45 min, washed, and cells were mounted for imaging using Fluoromount-G (Southern Biotech, Birmingham, AL, United States). For live labeling, cells were incubated with anti-FLAG antibody diluted 1:250 in serum-free media for 20 min in the culture incubator. Cells were washed, fixed with PFA, and immunostained as above. For tissue sections, mice were transcardially perfused with 4% PFA. Brains were post-fixed in PFA overnight and 50–100 µm vibratome sections were cut. Sections were incubated in blocking solution (PBS, 3% BSA, 0.2% Triton-X100) for more than 1 hr and incubated in primary antibody at 4°C overnight with gentle shaking. For VIP immunostaining, the blocking solution contained 0.3% triton +0.1% saponin. Secondary antibody incubation was done at room temperature for 2 hr. Sections were mounted in Fluoromount-G for imaging.

## Antibodies

Primary antibodies were used as follows: rabbit anti-Kirrel3 1:2000 (this study, generated against C-terminal peptide), rabbit anti-GABA 1:5000 (Sigma, St. Louis, MO, United States), goat anti-GFP 1:5000 (Abcam, Cambridge, MA, United States), guinea pig anti-VGLUT1 1:10,000 (Millipore, Billerica, MA, United States), mouse anti-MAGUK 1:1000 (NeuroMab, UC Davis/NIH NeuroMAB Facility, Davis, CA, United States), mouse anti-FLAG M2 1:5000 (Sigma), rabbit anti-synapsin 1:1000 (Millipore), rabbit anti-cFos 1:500 (Santa Cruz Biotech, Dallas, TX), rabbit anti-GFP 1:1000 (Invitrogen, Waltham, MA, United States), chick anti-FLAG 1:1000 (Gallus Immunotech, Cary, NC, United States), mouse anti-PSD95 1:2000 (Thermo Scientific), mouse anti-GAPDH 1:5000 (Millipore), chick anti-MAP2 1:10,000 (Abcam), rabbit anti-calretinin 1:2000 (Swant, Switzerland), mouse anti-parvalbumin 1:5000 (Swant), mouse anti-CamKII 1:5000 (Millipore), rat anti-Somatostatin 1:500 (Chemicon), rabbit anti-calbindin d28k 1:2000 (Swant), rabbit anti-VIP 1:500 (Immunostar, Hudson, WI, United States). All secondary antibodies were obtained from Jackson ImmunoResearch (West Grove, PA, United States) and used at 1:1000.

## $F_c$-binding assay

The extracellular domain of Kirrel3 was cloned in frame with human $F_c$ protein. Kirrel3-$F_c$ was transfected into HEK293 cells using PEI. Cells were incubated in OptiMEM (Life Technologies) media for 5 days. Then, the Kirrel3-$F_c$-conditioned media was harvested and concentrated using Amicon Ultra filter units (Millipore). Kirrel3-$F_c$ concentration was estimated by Western blot using known concentrations of purified $F_c$ (Jackson Immuno) as a standard. Kirrel3-$F_c$ was pre-clustered by incubating it in OptiMEM plus anti-human Cy3 secondary antibodies at 1:100. Kirrel3-$F_c$ and control $F_c$ (used at ~1 µg/ml) were then tested for binding to transfected HEK293 cells using the live label immunostaining method described above.

## Synaptosome preparation

Synaptosomes were prepared according to methods described by Jones and Matus with minor modifications (*Jones and Matus, 1974*). Briefly, hippocampi were dissected from mice. Tissue was homogenized with a Dounce homogenizer (20% wt/vol) in ice-cold 0.32 M sucrose +20 mM HEPES, pH 7.4 supplemented with protease inhibitors. Homogenates were cleared by spinning at 1000×$g$ for 10 min at 4°C. The supernatant was spun at 17,000×$g$ for 15 min. The pellet containing crude synaptosomes was resuspended in 0.32 M sucrose and 20 mM HEPES. Protein concentration was quantified with a BCA assay (Thermo Scientific). 2 µg of synaptosomal or cleared lysate proteins was loaded per lane for Western blot analysis.

## Western blot

Proteins were run on Bis-Tris gradient acrylamide gels and transferred to nitrocellulose membranes using the iBlot system (Life Technologies). Membranes were incubated in blocking solution (50 mM Tris pH 7.5, 300 mM NaCl, 3% wt/vol dry milk powder, and 0.05% Tween-20) for 10–60 min, primary antibody overnight at 4°C, washed, incubated in HRP-conjugated secondary antibodies (Jackson Immuno) for 1 hr at room temperature and then detected using the Bio-Rad Clarity ECL kit on a Bio-Rad ChemiDoc XRS+ imaging system. To prepare hippocampal lysates, 100 mg of hippocampal tissue was homogenized in 1 ml of reducing sample buffer.

## MF presynapse analysis

DiI crystals (Life Technologies) were placed in the DG of perfused hippocampi and incubated at 37°C in 2% PFA for 1 week. MF synapses from the suprapyramidal bundle in area CA3a/b were imaged, deconvolved in AutoQuant3 (Bitplane), and analyzed in ImageJ. Filopodia length was analyzed in 3D using the Simple Neurite Tracer plug-in, while area and perimeter of the main bouton were analyzed in 2D.

## Microiontophoresis and spine analysis

Neurons were microinjected with fluorescent dye as described (*Dumitriu et al., 2011*). Briefly, P21 pups were perfused with 1% PFA in phosphate buffer (PB) for 1 min, followed by 4% PFA with 0.125% glutaraldehyde in PB for 9 min. The brains were extracted and post-fixed in 4% PFA for 30

min. 200 µm-thick coronal hippocampal slices were cut on a vibratome. Slices were submerged in 0.1 M PB and viewed through an Olympus BX51WI microscope coupled to a light source and fluorescent filters. High-resistance (150–250 MΩ) glass pipettes were pulled on a Flaming/Brown P-97 Sutter pipette puller and backfilled with 10 mM Alexa568 (dissolved in 200 mM KCl, Life Technologies). The pipettes were mounted on a micromanipulator connected to an S44 Grass square pulse stimulator. The pipette tip was gently advanced in tissue towards the cell of interest. On contact and penetration, a step stimulus of 1–5 V was used to inject the dye in the cell. Filled neurons were imaged on a Zeiss LSM710 confocal microscope.

## cFos analysis

P14 and adult P60 wild-type and knockout mice were either removed from the home cage for immediate fixation by transcardial perfusion (unstimulated) or allowed to explore an enriched environment for 25 min prior to fixation (stimulated). The enriched environment consisted of mice placed individually into a $40 \times 40$ clear plastic box containing five novel objects spaced 10 cm apart. Mice were allowed to explore freely for 25 min. Immunostaining was conducted as described in the above methods.

## Image analysis and statistics

When possible, experiments were conducted by an experimenter blind to condition or genotype. Sample sizes were based on previous experiments or power analysis. Statistics were calculated in Prism (GraphPad). Intensity levels of some images were adjusted for visibility in publication but if so, the entire field of view and all comparable conditions were adjusted similarly. All images and conditions from the same experiment were collected and analyzed using the same confocal and analysis settings.

## Electrophysiology

Mice were rapidly decapitated and their brains carefully removed and kept in iced, artificial cerebrospinal fluid (aCSF) with sucrose (in mM—sucrose 200, KCl 3, $Na_2PO_4$ 1.4, $MgSO_4$ 3, $NaHCO_3$ 26, glucose 10, and $CaCl_2$ 0.5). 300-µm thick transverse slices were cut on a Leica vibratome (Leica VT1200) and the slices were left at room temperature in the holding chamber, until recording. P14–P16 mice were used for E/I ratio experiments and P17–P21 mice were used for mEPSC experiments. Neurons were visualized by differential interference contrast using a bright light source and an infrared filter on an Olympus BX51WI microscope with attached Hitachi color CCD camera (KP-D20BU). For mEPSC recordings: slices were continuously superfused with aCSF containing (in mM) NaCl 126, $NaHCO_3$ 26, KCl 3, $NaH_2PO_4$ 1.4, $CaCl_2$ 2.5, $MgSO_4$ 1, D-glucose 10, and TTX 1 bubbled with 95% $O_2$–5% $CO_2$. The intracellular pipette solution for mEPSC recordings contained (in mM) cesium methylsulfonate 80, CsCl 60, HEPES 10, EGTA 1 (adjusted with CsOH), $CaCl_2$ 0.5, glucose 10, and QX-314 5, adjusted to 290–300 mOsm/Lt at pH 7.3. For E/I ratio experiments, the aCSF was as above but without TTX and the intracellular solution was (in mM) cesium methylsulfonate 132, CsCl 8, HEPES 10, EGTA 1 (adjusted with CsOH), $CaCl_2$ 0.5, glucose 10, and QX-314 5, adjusted to 290–300 mOsm/Lt at pH 7.3. DCG-IV 0.5 µM (CAS no. 147782-19-2, TOCRIS) was perfused in the ACSF in some E/I ratio experiments to verify the specificity of stimulation. Unless noted, chemicals were sourced from Fisher Scientific (Pittsburgh, PA, United States).

Somatic whole-cell recordings were performed with Axon Multiclamp 700B amplifier (Molecular Devices, CA, United States) in voltage clamp mode at 34 ± 1°C bath temperature for mEPSC experiments and at room temperature (~22°C) for E/I ratio experiments. Data acquisition was performed via an Axon Digidata 1550 (Molecular Devices, CA, United States) with pClamp (Version 10, Molecular Devices, CA, United States). Current signals were sampled at 1 kHz and filtered with a 2-kHz Bessel filter. Patch pipettes with a tip resistance of 5–10 MΩ were pulled with a Flaming/Brown micropipette puller P-97 (Sutter Instruments and Co.) using borosilicate glass capillaries with filaments (1B150F-4, World Precision Instruments). Grass stimulator (S88, Grass Instruments) and bipolar tungsten electrodes (Harvard Apparatus, MA, United States, Ref. no. 72-0375) were used to deliver extracellular stimulation to the MF pathway at the hilus of the DG.

## Acknowledgements

We thank Hitoshi Sakano's laboratory for the Kirrel3 cDNA, Jean-Francois Cloutier for mice, Dimitri Tränkner for Matlab assistance, Jason Shepherd for manuscript comments, and the entire Williams lab. This work was funded by grants to MEW from the Whitehall, Alfred P Sloan, and Edward Mallinckrodt Jr Foundations, a University of Utah seed grant, and NIH grant 1R01MH105426.

## Additional information

### Funding

| Funder | Grant reference number | Author |
|---|---|---|
| National Institute of Mental Health | R01MH105426 | Megan E Williams |
| Whitehall Foundation | 3 year grant | Megan E Williams |
| Alfred P. Sloan Foundation | new investigator award | Megan E Williams |
| University of Utah | Seed grant | Megan E Williams |

The funders had no role in study design, data collection and interpretation, or the decision to submit the work for publication.

### Author contributions

EAM, Acquisition of data, Analysis and interpretation of data, Drafting or revising the article; SM, Acquisition of data, Analysis and interpretation of data, Drafting or revising the article; ZW, Acquisition of data, Analysis and interpretation of data; DCC, Acquisition of data, Analysis and interpretation of data; RB, Acquisition of data, Analysis and interpretation of data; MRT, Acquisition of data; JH, Acquisition of data; SAW, Acquisition of data; TC, Contributed unpublished essential data or reagents; AG, Conception and design; MEW, Conception and design, Acquisition of data, Analysis and interpretation of data, Drafting or revising the article

### Ethics

Animal experimentation: This study was performed in strict accordance with the recommendations in the Guide for the Care and Use of Laboratory Animals of the National Institutes of Health. All of the animals were handled according to an approved Institutional Animal Care and Use Committee (IACUC) protocol (14-07004) from the University of Utah.

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
