## [Decision Letter]

Thank you for submitting your work entitled “The intellectual disability gene Kirrel3 regulates target-specific mossy fiber synapse development in the hippocampus” for peer review at *eLife*. Your submission has been favorably evaluated by Gary Westbrook (Senior Editor), Graeme Davis (Reviewing Editor), and two reviewers, one of whom, Nick Spitzer, has agreed to reveal his identity.

The reviewers have discussed the reviews with one another and the Reviewing editor has drafted this decision to help you prepare a revised submission.

Martin, Muralidhar, and colleagues test whether the transmembrane Ig superfamily member Kirrel3 plays a role in mammalian synapse development. The authors hypothesize that perturbations of Kirrel3-dependent synapse targeting contributes to intellectual disability, as mutations of Kirrel3 have been associated with neurodevelopment disorders. The studies are well-designed and well-executed. The authors do a good job demonstrating that, in cultured hippocampal neurons, Kirrel3 is a hemophilic adhesion molecule and is present at the onset of synapse development, suggesting a similar role for the molecule in vivo. They provide evidence that other binding partners are not expressed in the same area of the hippocampus. They do a good job using anatomy to narrow their focus to DG and CA3, and provide convincing evidence that the absence of Kirrel3 disrupts the formation of MF filopodia.

Major criticism:

Intriguingly, the absence of Kirrel3 has an unexpectedly substantial impact on CA3 pyramidal cell activity in juveniles and on MF bouton morphology in adults. This is surprising because, according to the authors, Kirrel3 neurons represent a small subset of CA3 interneurons that are negative for common interneuron markers. This makes me wonder how such a small (and previously unknown) population could have a large impact in a developmental disorder. The findings would better illustrate the importance of Kirrel3 in hippocampal development (and might be important for subsequent electrophysiological and anatomical characterization of these neurons) if they included quantifications of the density of Kirrel3-positive interneurons in this area, the fractions of GABA-positive neurons in CA3 that are Kirrel3-positive compared to the fractions of other interneurons, and whether these fractions are unchanged in the mutant (using GFP as a marker).

Minor criticisms:

The authors claim that the change in MF bouton morphology indicates a homeostatic compensation for CA3 hyper excitability, but do not supply corresponding excitatory/inhibitory experiments to test whether this parameter has been corrected. Such experiments would be an important contribution toward understanding the role of Kirrel3 in neurodevelopment disorders.

The authors are unable to identify Kirrel3-positive neurons in wild-type mice, rendering many experiments – including direct examination of inhibitory neurotransmission at the MF-GABA synapse, or anatomical studies of the direct interaction of filopodia and Kirrel3-positive neurons – difficult to do. This might be ameliorated by improvement in the approach to labelling neurons in the wild-type, either by improving the antibody (over what is shown in Figure 4), the staining technique, or using a commercially available antibody against Kirrel3.

Please describe the mice used in the study either in the Results or Methods section (or at least refer to the paper if these are the same farnesylated GFP mice generated previously, Prince et al., 2013).

Please be consistent throughout the Results and Discussion sections in listing the ages of the mice tested.

Please detail the enrichment protocol you used for the cFos induction part of the study.

Please include representative images of adult MFB.

In order to tie this result to the effect of neurodevelopmental disorders throughout the brain, can you expand on your point in the final paragraph, on how the proper formation of specific synapses might contribute to functional defects of other brain regions?

In the subsection “Kirrel3 regulates MF filopodia development”, are the filopodia whose lengths are measured at P14 all presynaptic, or are some of them still growing toward the GABAergic interneurons?

In the subsection “Kirrel3 regulates CA3 neuron excitability driven by MF input” a citation is needed to document the selective inhibition of MF release by mGluR activation.

In the subsection “MF presynapse analysis”, why was filopodial length analyzed in 3D and the perimeter of the main bouton analyzed in 2D? 3D would seem to be more accurate.

---

## [Author Response]

Major criticism:

Intriguingly, the absence of Kirrel3 has an unexpectedly substantial impact on CA3 pyramidal cell activity in juveniles and on MF bouton morphology in adults. This is surprising because, according to the authors, Kirrel3 neurons represent a small subset of CA3 interneurons that are negative for common interneuron markers. This makes me wonder how such a small (and previously unknown) population could have a large impact in a developmental disorder. The findings would better illustrate the importance of Kirrel3 in hippocampal development (and might be important for subsequent electrophysiological and anatomical characterization of these neurons) if they included quantifications of the density of Kirrel3-positive interneurons in this area, the fractions of GABA-positive neurons in CA3 that are Kirrel3-positive compared to the fractions of other interneurons, and whether these fractions are unchanged in the mutant (using GFP as a marker).

We thank the reviewers for their time and thoughtful comments. In the revised manuscript, we conducted a more thorough and quantitative analysis of the molecular identity of Kirrel3-positive interneurons. Because we still lack a suitable Kirrel3 antibody to routinely identify Kirrel3-expressing cells in wildtype mouse tissue, we conducted our analysis in P14 heterozygous and knockout mice, which have one or two copies respectively of GFP knocked in the Kirrel3 locus. We used GFP to identify Kirrel3-positive cells and made several new discoveries that are detailed in revised Figure 2 and Figure 2—figure supplement 2. First, nearly all Kirrel3-positive cells (excluding DG neurons) are in fact GABAergic (over 90%). Second, we included more GABAergic markers and discovered most Kirrel3 GABA neurons (67%) express calbindin-D-28K. A smaller fraction (<20%) is somatostatin positive (the two markers are not always mutually exclusive). There is very little overlap with VIP, parvalbumin, or calretinin. Moreover, ∼20% of all GABA neurons in area CA3 express Kirrel3. These cell populations are very similar in P14 Kirrel3 heterozygous and knockout mice. This suggests loss of Kirrel3 does not change cell fate or induce cell death of the neurons examined and that wildtype mice likely have similar numbers. The role of calbindin-positive GABA neurons in the hippocampus is little studied but our new results suggest they may receive a substantial portion of DG mossy fiber input and are likely a critical component of feed forward inhibitory circuits that constrain CA3 activity.

Minor criticisms:

The authors claim that the change in MF bouton morphology indicates a homeostatic compensation for CA3 hyper excitability, but do not supply corresponding excitatory/inhibitory experiments to test whether this parameter has been corrected. Such experiments would be an important contribution toward understanding the role of Kirrel3 in neurodevelopment disorders.

We agree measuring E/I ratios of adult CA3 neurons after MF stimulation would provide additional data to test our hypothesis that adult mice have undergone homeostatic changes to correct CA3 hyper-excitability. Unfortunately, whole cell recording from adult CA3 hippocampal neurons is rarely reported because of the technical difficulty in obtaining quality recordings from healthy cells in adult tissue. CA3 neurons are particularly difficult and among the first hippocampal cells to undergo cell death in acute slices. We understand this is not an impossible experiment, but we hope the reviewers appreciate it is not trivial and has only been accomplished by a few labs with deep electrophysiology experience. After several months, we obtained some E/I recordings from P40-50 slices, but each time DCG-IV was added to confirm MF axon stimulation the cells were lost. They generally support our conclusion but we excluded them from the manuscript because of our inability to confirm they directly resulted from MF stimulation. Nonetheless, our cFos analysis in adult mice (Figure 4) supports our hypothesis and has led to our working model that will be tested in future studies. We revised the results and discussion to make it clear that the adult compensation effect is still a working model resulting from the many novel findings presented in this Short Report.

*The authors are unable to identify Kirrel3-positive neurons in wild-type mice, rendering many experiments – including direct examination of inhibitory neurotransmission at the MF-GABA synapse, or anatomical studies of the direct interaction of filopodia and Kirrel3-positive neurons – difficult to do. This might be ameliorated by improvement in the approach to labelling neurons in the wild-type, either by improving the antibody (over what is shown in*Figure 4*), the staining technique, or using a commercially available antibody against Kirrel3.*

We agree identifying Kirrel3-positive interneurons in wildtype mice is important for future studies of Kirrel3 function in the brain. We have exhaustively tested alternative conditions to optimize the antibody shown in Figure 2 (including changing detergents, fixation times, cryosectioning, and antigen retrieval methods) with no labeling improvement. Moreover, we tested antibodies from 5 commercial vendors and have not yet identified another antibody that specifically labels Kirrel3 in wildtype, but not knockout, mice by immunohistochemistry. Now that we discovered most Kirrel3-GABA neurons express calbindin, we are actively pursuing intersectional viral and transgenic strategies to label Kirrel3-GABA cells in wildtype and knockout mice. However, this is a long-term, on-going project that we hope the reviewers agree is outside the scope of this study.

*Please describe the mice used in the study either in the Results or Methods section (or at least refer to the paper if these are the same farnesylated GFP mice generated previously,Prince et al., 2013*).

Kirrel3 knockout mice used in this study are the same mice as described previously in Prince et al. 2013. We clarified this in both the Results and Methods sections.

Please be consistent throughout the Results and Discussion sections in listing the ages of the mice tested.

The ages of animals used in each experiment are clearly stated in the revised text and figure legends.

Please detail the enrichment protocol you used for the cFos induction part of the study.

We apologize for omitting this in the original submission. It is included in the revised submission.

Please include representative images of adult MFB.

Representative images of adult MFBs are now included in the revised manuscript Figure 4.

In order to tie this result to the effect of neurodevelopmental disorders throughout the brain, can you expand on your point in the final paragraph, on how the proper formation of specific synapses might contribute to functional defects of other brain regions?

We expanded the Discussion to expand on the role of Kirrel3 in disease and discuss our new findings.

In the subsection “Kirrel3 regulates MF filopodia development”, are the filopodia whose lengths are measured at P14 all presynaptic, or are some of them still growing toward the GABAergic interneurons?

Currently our methods label DG axon membranes, not synapses directly (we clarified this in the revised manuscript). As such, we cannot conclude if the identified filopodia end in synaptic contacts or not. Elucidating this will require electron microscopy. We are currently conducting a follow-up study using electron microscopy to more precisely analyze filopodial synaptic defects in Kirrel3 knockout mice.

In the subsection “Kirrel3 regulates CA3 neuron excitability driven by MF input” a citation is needed to document the selective inhibition of MF release by mGluR activation.

Citations supporting the role of DCG-IV in selective inhibition of MF release were added to the revised manuscript.

In the subsection “MF presynapse analysis”, why was filopodial length analyzed in 3D and the perimeter of the main bouton analyzed in 2D? 3D would seem to be more accurate.

We initially quantified the main bouton using 2D area and perimeter measurements (as reported) and 3D volumetric measurements, both by an experimenter blind to genotype. Though the 3D bouton analysis supports our hypothesis that the size of the main bouton is similar between WT and KO mice at P14 (WT = 25.1μm^2^ ±12.27 st. dev. versus KO = 22.2μm^2^ ±11.41, p=0.25 by t-test), we excluded these results from the manuscript because we find analyzing the main bouton volume by light microscopy is inaccurate. It tends to be 2-3 times larger than volumes reported by electron microscopy. MF boutons are highly irregular, sometimes hollow, shapes molded to multi-headed thorny excrescence spines. Analyzing these complex shapes by light microscopy in 3D is hampered by the resolution of confocal images in the Z direction and image analysis software tends to recognize these boutons as solid rounded shapes, artificially increasing the volume. In 2D, we can easily draw irregular shapes corresponding to z-projected MF bouton area and perimeter. It is more accurate to model filopodia length in 3D because it requires only quantification of lines. We assume filopodial volumes would also be inaccurate and this was not analyzed. Our values for WT filopodia number and length closely match previously reported values determined by electron microscopy. Regardless, all WT and KO images were similarly analyzed by an experimenter blind to genotype and are comparable.